# Mutual Capacity Building through North-South Collaboration Using Challenge-Driven Education

**Anna-Karin Högfeldt** [1],*, **Anders Rosén** [2], **Christine Mwase** [3], **Ann Lantz** [4], **Lena Gumaelius** [1], **Eva Shayo** [3], **Suzan Lujara** [3] and **Nerey Mvungi** [3]

1  Department of Learning in Engineering Sciences/School of Industrial Engineering and Management, KTH Royal Institute of Technology, 100 44 Stockholm, Sweden; lenagu@kth.se
2  KTH Global Development Hub, KTH Royal Institute of Technology, 100 44 Stockholm, Sweden; aro@kth.se
3  College of Information and Communication Technologies, University of Dar es Salaam, P.O. BOX 35091 Dar es Salaam, Tanzania; cmwase@ieee.org (C.M.); eva.shayo@gmail.com (E.S.); suzyluj@gmail.com (S.L.); nhmvungi@gmail.com (N.M.)
4  Division of Media Technology and Interaction Design/School of Electrical Engineering and Computer Science, KTH Royal Institute of Technology, 10044 Stockholm, Sweden; alz@kth.se
*  Correspondence: akhog@kth.se

**Abstract:** The urgent need for actions in the light of the global challenges motivates international policy to define roadmaps for education on all levels to step forward and contribute with new knowledge and competencies. Challenge-Driven Education (CDE) is described as an education for Sustainable Development (ESD) approach, which aims to prepare students to work with global challenges and to bring value to society by direct impact. This paper describes, evaluates and discusses a three-year participatory implementation project of Challenge-driven education (CDE) within the engineering education at the University of Dar es Salam, UDSM, which has been carried out in collaboration with the Royal Institute of Technology, KTH in Stockholm. Conclusions are drawn on crucial aspects for engineering education change through the lens of Activity Theory (AT), where CDE is brought forward as a motivating ESD initiative for engineering faculty and students. Furthermore participatory co-creation is notably useful as it aims to embrace social values among the participants. Also, traditional organizational structures will need to be continuously negotiated in the light of the integration of more open-ended approaches in education.

**Keywords:** ESD; challenge-driven education; engineering education; higher education change; participatory action research; activity theory; challenge based learning

## 1. Introduction

Through the adoption of the UN's 2030 Agenda, the global society and governments all over the world have agreed on the urgent need for change [1]. The Sustainable Development Goals (SDG), that constitutes the core of the Agenda, formulates a shared view of the global challenges that are crucial for humanity in the 21st century. The 2030 Agenda and the SDGs can be claimed to induce a shift in the world logics. According to the old logic, for example represented by the Millennium Development Goals (MDG) that were preceding the SDGs during the period 2000–2015, the world was divided in developed countries and developing countries, where the developing were to transform towards the developed. However, in light of the 2030 Agenda and the SDGs, we are all to be considered as developing countries in need for substantial transformations [2].

The importance of education and the crucial role of higher education institutions for achieving sustainable development is regarded by many as obvious, e.g., [3–5] and one of the SDGs focus

particularly on education. Education does however not contribute to sustainable development per se. Education that promotes economic growth alone may for example as well promote unsustainable consumption patterns. Education for Sustainable Development (ESD), by some called a "global movement" [6] (p.752) has been analysed in several works, for instance in [7,8]. UNESCO [9] (p. 7) outlines the following important features of ESD:

- ESD should integrate sustainable development concepts and content such as climate change, poverty, sustainable consumption, etc, into the curriculum;
- ESD should develop key competencies for sustainability that empowers learners to take informed decisions and responsible actions in complex situations for environmental integrity, social justice, and economic viability, for present and future generations from a local as well as global perspective;
- ESD should truly matter and be relevant to the learners in the light of today's challenges.

  UNESCO [9] (pp. 47–57) further characterizes pedagogical approaches adequate for ESD as:

- Learner-centred and seeing students as autonomous learners actively developing and reflecting on their knowledge and competencies;
- Action-oriented and engaging learners in real world contexts and situations that promotes linking abstract concepts to personal experiences;
- Transformative and aiming at empowering learners to question and change the ways they see and think about the world, or even transgressive aiming at preparing the learners for disruptive thinking and co-creation of new knowledge;
- Multi-perspective and involving students with different disciplinary and cultural backgrounds, and a range of societal actors such as businesses, NGOs, public institutions, policymakers, etc.

This paper considers the development of a partnership and a challenge-driven pedagogical approach that implements the ESD features and related pedagogical characteristics outlined by UNESCO [9] in the curriculum of five universities in Botswana, Kenya, Rwanda, Sweden and Tanzania; they are all partners in the KTH Global Development Hub network. The intention with KTH Global Development Hub is to create conditions for students, faculty and various societal actors from the involved universities and countries, to collaborate on education, research and innovation, and mutual capacity building to promote sustainable societal transformations in the South as well as in the North. Part of this endeavour is to collaboratively develop and implement challenge-driven education (CDE) at all involved universities and establish a student exchange program that enables composition of teams of students from the different universities.

Challenge-driven education is a concept in evolution. Various examples of implementation can be found on the primary and secondary levels of education [10] as well as in higher education [11–16]. CDE implemented in higher education resembles real problem based learning as defined by Kolmos et al. [17], for example in that the learning is built around open ended projects where the development of solutions requires knowledge and skills beyond that of a single discipline. CDE is being implemented somewhat differently by different teachers at different universities.

The challenge-driven education concept CDE$^{GDH}$ that is developed and promoted by KTH Global Development Hub (GDH), can be described as a solutions- and impact-oriented project-based approach where multi-perspective student teams collaborate with various external stakeholders in projects that are addressing societal challenges related to the Sustainable Development Goals (SDGs) in UN's 2030 Agenda. CDE$^{GDH}$ provides opportunities for students to develop key competencies for sustainability, to deepen their disciplinary knowledge and professional skills in realistic contexts, and to create value for the society already during their studies. CDE$^{GDH}$ provides opportunities for universities to enhance outreach and interaction with the society and involved external stakeholders get access to the knowledge and creative force of students and universities. CDE$^{GDH}$ projects involve teams of students with different disciplinary backgrounds and preferably also different geographical and cultural backgrounds. This promotes inter/trans-disciplinary/cultural learning and enables broader scopes for

the projects and challenges to be addressed. CDE^GDH projects engage various external stakeholders such as companies, public sectors, communities, NGOs, end users and others who can affect or be affected by the project and its outcomes. Some of these can take more active roles in identifying or providing challenges and in receiving and further exploiting the project results. CDE^GDH aims for making actual impact on the sustainable development of the society through innovation, societal capacity building, and by students becoming change agents for sustainable transformations.

The paper focuses on the application of a participatory action based research (PAR) approach for developing and implementing CDE at the University of Dar es Salaam (UDSM) in Tanzania in collaboration with the Swedish university KTH Royal Institute of Technology. The project has been ongoing during the years 2016–2019. Section 2 provides the rationale for applying participatory approaches in the implementation of CDE. In the following methodology section, the PAR methodology, research scope and methods used will be described together with the Activity Theory framework which is applied in the analysis of the findings. The Section 4 provides the data and materials for the two implementation phases from year 2016–2018 and 2018–2019 respectively. The first phase was previously described in [15]. Summative findings of the perceptions among the key stakeholders, namely the students, the teachers and the challenge providers, are presented in Section 5, and analysed from an Activity Theory framework in Section 6. The study is discussed in Section 7 and conclusions are presented in the Section 8. An overview is provided in Figure 1.

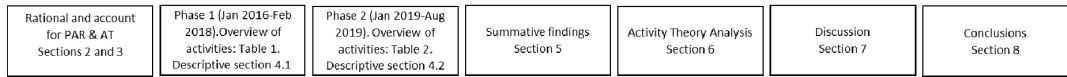

**Figure 1.** Study overview of the Participatory Action Research project.

In the light of the collectively gained experiences of the CDE implementation among teachers, students and societal stakeholders, the overall question for the study targets what we can learn about participatory education change from an activity system perspective, when integrating CDE in engineering education, and how this can inform future steps for the integration of CDE.

## 2. The Call for Participatory Approaches in Higher Education Change towards ESD and CDE

Tanzania and Sweden have long traditions of collaboration. Since at least half a century, both among universities as well as other organizations, various projects have been accomplished. This is true also for the University of Dar es Salam (UDSM) and the KTH Royal Institute of Technology. Previous collaborations between the two have been targeting research, development and PhD education. In 2014 the idea emerged to also collaborate on first and second cycle education, with the vision of mutual capacity building through joint development and implementation of challenge-driven education (CDE). The idea was to provide students from the two universities opportunities to collaborate with each other and societal stakeholders on developing solutions to various societal challenges in the two countries. The teachers at UDSM expressed that they would like to hear from KTH teachers about their previous experiences of CDE, and a Guide for Challenge-Driven Education was written through funding from KTH [11]. The guide includes cases based on interviews with teachers, insights in different student projects and pedagogical approaches for project based teaching. While the guide was perceived as handy and practical, it was clear that this type of information transfer would not make the vision realized. Further, the ongoing evolution of the CDE concept quite soon made the printed guide somewhat outdated. All involved, teachers, students, educational developers and researchers as well as external challenge providers, in both countries, would instead need to establish more deeply rooted methods and processes for the educational change to happen and become durable. Such approach is further supported by an OECD study on the management of change in higher education institutions in Tanzania [18] (p. 138) that reveals a "directive leadership". They underline "the importance of participatory management, which demonstrates the need to promote autonomy,

encourage teamwork and secure people's motivation and commitment to tasks and thereby obtain their best performance" [18] (p. 140).

## 2.1. Higher Education Change Models

The trends among the emerging leaders of engineering education are to have clearer links to the society and the regional development; crossing disciplines in innovative ways; focusing on engineering design as well as the student's self-awareness [19]. In order for change to happen, a common initiation is a significant 'threat' from the outside [20] (p. 2). This threat, or rational, can come from perceived or factual pressures from similar universities, the state, higher education arenas or "globally circulated ideas" [21] (p. 1). Normally it's not an isolated event where only a few are involved, and for change to happen, Department Heads or similar have to be involved and committed to the change. Furthermore, [20] (pp. 10–11) lists 'critical features of success and failure' which are: Leadership, communication and vision; faculty development; faculty engagement; resources and time; external networks; culture and rewards procedure; and, finally sustaining the change through monitoring, giving structural support, information to new staff on what and why and the establishment of an on-going focus on education.

Processes for change management of higher education can be carried out in many ways and the most commonly used dichotomy to explain change is perhaps the one which targets the bottom-up vs top-down. Kezar [22] provides a more thorough and substantial framework for approaches to change. The argument is made that when analysing change management procedures one needs to look at the why, what, how and the target of change [22]. There can be many reasons, or forces and sources of change. Change can concern a superficial layer, first-order, or can target deeper change levels, second-order, involving for instance attitudes and other cultural aspects. Change can be carried out by a few, or by many. It can be adaptive or generative; proactive or reactive; planned or unplanned [22].

There are a handful of main approaches, or schools, for the management of higher education change, explained in [22] which will combine the why, what, how and target in different ways. According to [22] several of them happen at the same time, or several of them can be applied when planning for change. The scientific management models, where a rational and linear change is planned internally at the institution, are often carried out with a strong steering committee, and often lacks human emphasis. In a review study on successful quality cultures in higher education institutions [23], the scientific management approaches reveal little if any impact on change on its own. Högfeldt et al. [24] conclude in a study among educational leaders within five Nordic countries that the level of 'informal power', or 'social power' [25] (p.334) a leader perceives having, will influence his/her strategies for change. In the evolutionary models, the external environment is the driver for change, the change comes slowly and sometimes non-intentionally [22]. During the last decades, there has been an increased awareness of and influence by managerialism [21,26], organizational barriers [27], with its related agenda setting, bargaining and negotiations [22,28,29] which points to the importance of analysing the reasons for change with political models [22]. Two main models looking more at how change comes about rather than why change is initiated are social cognition and cultural models. In social cognition models, change is connected to mental processes among people involved, since their learning and understanding is important for change. Interpretations and sense-making is more important than rational explanations. This school brings in the idea of faculty training as a key to change [30], but was in its early days criticized for not taking into account feelings and values among the stakeholders of change [22]. This has influenced the emergence of cultural models, which bring in multiple layers of values, rituals and attitudes, see for instance [31,32]. Bendermacher et al. [23], in their review on which strategies actually lead to successful change, propose a holistic quality culture system where inclusive and collaborative approaches, with room for bottom-up ideas and ad-hoc initiatives from empowered faculty, staff and students are embedded in a system with strong leadership and articulated common visions [23].

## 2.2. Implementation of Sustainable Development in Higher Education

As for the implementation of sustainable development in higher education, the approaches to change are diverse, and are aimed at different layers in the organization [33–35]. In the literature on SD implementation in higher education the conclusion that rational or evolutionary models on their own will not make durable change is prevailing. Rather, social cognition and cultural models should be developed and applied, with an aim to create holistic and participatory processes [6,36–38].

Verhulst and Lambrechts [34] have developed a conceptual model, based on the work by [33,39–41], where four key issues for change in the integration of SD are considered which are related to human factors. The first issue to consider is the resistance to change, and to understand that not all resistance stems from demotivated faculty. Rather, the emphasis on research excellence and the lack of time for education development stand out as major barriers to consider. The second aspect related to the implementation of SD is according to [34] the communication of the changes. In several of the studies reviewed by [34] there has been a critique over the implementation processes where the communication has been perceived as too general and infrequent. The third factor is the empowerment and involvement of the stakeholders who will be affected by the implementation of SD. People involved should have authority, resources and specialisation skills as well as self-determination. Finally, the fourth condition for the implementation of SD is to consider the organizational culture. The whole institution needs to adapt to the values of SD if the implementation should be long-lasting [33].

The actors who carry the heaviest burden in the implementation of SD in higher education teaching and learning activities, could be argued to be the teachers who are responsible for the execution and implementation. According to [8] university faculty world-wide have little support and training for the work which the SD implementation processes imply. The need for ESD training among faculty is crucial [42]. Mulà et al. [8] highlights crucial aspects for professional faculty and teachers in relation to ESD which has been formulated by the University Educators for Sustainable Development in 2016 [43]. Examples on the expectations of higher education teachers are the abilities to "rethinking assessment of student progress and achievements"; "challenging power relationships in learning and engaging students at all levels of the learning dynamic" and the ability to "digesting how sustainability thinking and practice articulates in different industries/professions" [43] (p.1). Mulà et al. [8] finds that on an institutional level, actively involved faculty in the change processes towards SD are often only those individuals, or academics who already are researching in the field of sustainability. In order to support a broader part of the higher education faculty to become part of the change towards ESD, Mulà et al. [8] conclude that there should be a co-developed professional development program among experts and newcomers, with mentoring as a core activity. Educators should be supported to transform education, in the development, practice and assessment, rather than only be given conceptual lectures on SD in education, a viewpoint generally shared by many in the higher education teacher training sector [44].

## 3. Methodology

In this section the foundations of the framework for the implementation of CDE at UDSM, the Participatory Action Research (PAR) is described, including the scope of the research and the methods applied. Furthermore, the Activity Theory (AT) framework for the analysis of the findings is described.

### 3.1. Particpatory Action Research

Disterheft et al. [6] argue that participatory approaches can be successful in SD implementation. Participatory Action Research (PAR) targets practical problems which are examined and handled [45]. PAR can be defined as "a process of investigating, understanding, reflecting upon, establishing, developing, and supporting mutual learning between multiple participants in collective reflection-in-action" [46] (p.21). The research process should lead to progress towards a shared goal, provide with change and new experiences, and not merely have the aim to be checking out

or monitoring a process [45,47,48]. Due to its practical nature, the limits with PAR can be a lack of generalizability, although lessons learned are often transferable while taking into account contextual differences [45].

PAR emerged during the 1960s and 1970s. This movement away from positivist and imperial views has its roots in the work by [49–51], to name a few but important researchers in the fields of social and educational sciences. Research should according to their views be contextualized by participation and action and break out from "traditional methods, with the emphasis on step-by-step procedures, which effectively prevent creative and cooperative sparks between system designers and users" [52] (p. 8). Lawson et al. [53] position PAR as a political process, since there is a goal set up that should be achieved, and there is a political charge related to the selection of the participants.

The view of the researcher(s) within PAR is different from when a conventional researcher works with a study from an outside perspective [48]. In PAR, the researcher(s) should be actively involved in the project [45]. The researcher(s) are neither positioned ahead of or above other participants in the project [45,47,54].

The knowledge needed for continuous development and change of education is seen to be gained through shared learning and collaboration among all stakeholders involved [45,53,55]. High autonomy should be given to the team of teachers who will be working closest to the new setting [56]. It is their socially constructed sense-making which is the engine for the PAR process [57]. By using their gained and accumulated knowledge, the results of the process is regarded to be more durable [46]. Therefore, active involvement should be encouraged, and development should be carried out grounded in the shared experiences of all stakeholders [47], embracing the sociocultural perspectives on the realization of change through learning and development in communities of practice [45,58,59].

The actual set-up of PAR should be contextually and continuously developed, where methods for implementation and inquiry are user-oriented [45]. Arenas should be provided for all voices to be heard, and for people to meet, listen and learn from each other [47]. Often, there will be prototyping and piloting processes, in a design-by-doing format [46]. Written reflections like diaries or portfolios can also be valuable tools to invite all participants to contribute with their experiences [47]. PAR should be systematically designed in cycles where reflection and action alternate [45].

Result validation in PAR should also actively invite all participants' views, where the target is to understand if practice has been improved towards the desired shared goal [45]. This should not end with only asking the participants on their perceptions and views, but to create arenas for all to be actively involved in the sketching of the coming steps [46], leading to social change [60]. Throughout the process, transparency is key in order to create and maintain trust and motivation among the participants [54].

Disterheft et al. [6] have in their work suggested four key factors to contextualize participatory processes in the implementation of SD in higher education. Firstly, the level of engagement should be high and aim at engagement in decision-making and empowerment. Secondly, a broad diversity of stakeholders should be involved, such as students, teachers, and other staff members at the institution as well as external stakeholders. Thirdly, specific quality aspects of the participatory processes should be considered—stimulating systems thinking, critical thinking and reflections about values; providing spaces for negotiation of goals and outputs; analyzing the level of satisfaction of participants; and sharing existing and generating new knowledge. Finally, according to [6] new cycles of participation should involve the learning from previous cycles and the new gained knowledge and values should preferably be shared. In [36] it is argued that the implementation of SD should be of a learning and process oriented approach, rather than only measuring the output of SD education. Kezar [61] argues that in the work with participatory change and leadership there are pitfalls that will need to be avoided. There is a risk to be unaware of different power relationships. Also, one might be unobservant of outcomes that can be oppressive. A dominant model might emerge that participants is expected to accept, where little room for negotiation is given and individuals are forced to align [61].

*3.2. Research Scope and Methods for the Implementation Project of CDE*

In this study, the aim is to create a 'collective reflection-in-action' [46] (p.21). The research should lead to a transformation of engineering education at UDSM and KTH. In the desired scenario [47], both universities will be conducting challenge-driven education in collaboration, sharing experiences and providing opportunities for students to participate in each other's CDE courses. In line with the idea to contextualize the processes by participation in action [52], a team was set up consisting of teachers, university leaders from both universities, as well as educational developers and researchers at KTH [45]. The scope for the participatory action research team was to join and support the educational transformation through the years of 2016–2019, in order to support the educators in the development, practice and assessment [8,45].

During the first phase, from 2016–2018, a pilot challenge-driven education course on MSc and PhD level was implemented and tested locally at the College of ICT (CoICT) at UDSM. This work has previously been presented at two conferences [15,62]. In the second phase, a BSc course was implemented inviting students not only from the CoICT, but also from the department of water resources engineering, as well as students from the corresponding departments at KTH.

Forums for planning and evaluation through the project were designed along the way, avoiding step-by-step procedures [52] in order to promote creativity, participation and mutual learning [57] that would guide the most reasonable coming steps [6]. The methods have been chosen based on the needs or issues which the CDE implementation team have discovered [45]. A variety of arenas for reflections have been created as opportunities have emerged [48]. The results from the continuous reflective activities such as the workshops, the group interviews, the mid-term presentations, as well as the gained knowledge and decisions on actions between the phases, are described in Section 4.

To finalize the first phase, it was decided during the project to send out an online questionnaire in July 2017 to the students, the teachers and the external stakeholders who had been involved in the pilot CDE project during the first phase of the implementation process. For the students and the teachers, the questions focused on four different perspectives: curriculum, project-based learning, challenge providers and course design. The questions were mainly open and had no limits of space. For the challenge providers, the questionnaire included mainly open questions on the meetings and discussions they had been involved in with the students and the teachers, how the idea with CDE and the projects were perceived, how well students could grasp the challenge and the perceived value for their company. Replies were received by all 15 students (S1–S15), all four teachers (T1–T4) and 8 of the 12 challenge providers (C1–C8).

After the second phase, in July 2019, the participants gathered in Stockholm in August 2019 on a reflective seminar week, to share their experiences from the two phases, evaluate the conceptions of CDE, and discuss the participatory action research methods as well as designing a road map for the future. The students and teachers from both universities were asked to prepare the contents of the seminar, by summarizing their reflections and evaluations of CDE which they presented orally at the workshop. All preparations were recorded, transcribed and thematically summarized. Furthermore an online questionnaire was open for all workshop participants, in order to invite all to share their experiences also anonymously if needed. The questionnaire included three open questions. The first two questions concerned the benefits as well as the improvement needs with the challenge-driven education approach. The third question focused on the participants' perceptions on how insights on sustainability and sustainable development goals can be gained, or not, through CDE. One student from UDSM, four teachers from KTH ($T_21$–$T_24$) as well as three teachers from UDSM ($T_25$–$T_27$) used the online questionnaire to share their experiences. These responses were merged with the summarized oral reflective presentations.

The study has been conducted participatory and is co-authored by the involved teachers, the developers and the researchers in the project in order to realize the idea with PAR where the researchers are not external from the project [45,48,54]. The selection of team members have been based

on previously established contacts, where the involved teachers could be argued to be committed to the change initiatives [53].

### 3.3. Activity Theory as a Framework for Analysis

The implementation of challenge driven education in engineering education is seen as difficult, and mainly ends up as an extra-curricular activity due to its complex nature of a radically different teaching and learning set-up [12]. Therefor Activity Theory (AT) has been applied to analyze the summative results of the perceptions among the students, the teachers and the challenge providers of the CDE implementation [63,64]. AT studies change from a system perspective and is seen as a fruitful analysis framework for action oriented research [60]. What the applications of AT in the field of education can be argued to have in common is the study of subjects who are part of a system which is changing towards something new, the object of the change [64,65]. Study topics within the field of education extends from research in educational technology [66], such as the analysis of learning activity patterns in learning management systems [65], to science-teachers' professional learning [67] and the implementation of student-centred learning in traditional education [68] as well as to comparative studies of different forms of PhD educations [69]. AT can also be found in research studies in the fields of psychology, technology as well as work and organizational related studies [70].

The activity theory (AT) was founded by the psychologist Vygotsky in the 1920's and further developed by Leont'ev and their respective teams [70,71]. Engeström [63] has generated the theories of expansive learning from the AT [69]. The activity system is normally drawn as a triangle with key components, see Figure 2. The subject(s) is the person or group who will be influenced by the changes [67], and it's from the subject's perspective the AT analysis is made [66]. An object is something "radically new, wider and more complex" phenomenon [72] (p. 2) than the regular activities in the system, which is implemented, directing the activities and actions among the subject [67].

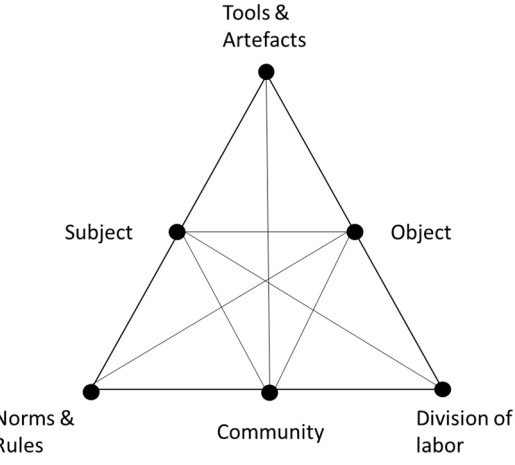

**Figure 2.** "The structure of human activity system" [63,64].

While mediating artefacts are used and developed in order to reach the desired scenario and make sense of the object, this is only the "tip of the iceberg" [64] (p.47). The activities in the system are collective and therefor Engeström [64], based on the work by Leont'ev [71] expanded the visualization of the system to include the explicit and implicit norms and rules, the relations to the community and the "social context" [69] (p.991) as well as how labour is distributed and re-distributed with the activities [64,67]. Murphy & Rodriguez-Manzanares [66] argue that an important benefit with AT is that it opens up for the broad and historical perspective on stakeholders' views and the settings around them. It can bring light to the understanding of how actions are motivated [65] and provide with a focus on the analysis of the contradictions in the system and opportunities to revealing the barriers [65,66]. By this, contradictions can be solved "through the generation and integration of new practices in the

activity system, and through the redefinition of its elements" [69] (p. 992). Actual change is argued to happen when contradictions within the system are solved [64,65].

In this study the activity system is 'engineering education' in a simplified form, where the selected subjects, whose perceptions are analyzed, are the groups of students, teachers and challenge providers involved in the PAR implementation. The object is the challenge-driven education, with the desired outcomes to reach an optimal combination of education, research and collaboration as well as to educate future change agents for a sustainable society. In the analysis, the focus is on the main contradictions, or rather the "ambiguity, surprise, interpretation, sense-making, and potential for change" [64] (p.47) which the subjects face, and the actions made in relation to the components of the activity system. When developing and using different mediating artefacts, norms, communities and divisions of labor in order to make meaning of the object, what does this say about the change processes in the implementation of challenge-driven education?

In the analysis, the first step was to code each subject groups' summative perceptions over the two implementation phases with regards to aspects related to tools, norms, community as well as the division of labor. For instance, the use of online instructions to improve learning across the nations, was labelled as a mediating artefact, while students stating they would walk the extra mile, was labelled as a division of labor. When all three groups' perceptions were coded, three activity system triangles were sketched next to each other. After the labels and codes were organized for each of the subject groups, all the three triangles were analyzed together by looking at one relation at a time – the subject's relation to the artefacts, to the norms etc–on each of the three triangles. The guiding question in this analysis was: what is the essence of the three subject groups' perceived relation to each of the respective component in the activity system [73]? The overall aim with the analysis is to understand what we can learn about participatory education change from an activity system perspective when integrating CDE in engineering education and how this can inform future steps for the integration of CDE.

## 4. Data and Materials of the Participatory Implementation of Challenge-Driven Education at UDSM

This section provides with details on the participatory implementation processes throughout the years of 2016–2019. Important milestones are shared, which are connected to Table 1, for the first implementation phase and Table 2 for the second.

**Table 1.** Overview of the participatory activities that were conducted in the first phase.

| What | When | Where | Who |
|---|---|---|---|
| Planning workshop. | Aug. 2016 | DeS | Project team teachers (KTH, UDSM, DIT) |
| Challenge definition workshop. | Sept. 2016 | DeS | UDSM teachers and students & 3 TANESCO partners |
| Course start. | Oct. 2016 | UDSM, CoICT | UDSM teachers and students |
| Evaluation and planning workshop. | Dec. 2016 | KTH | Project team (KTH, UDSM, DIT) |
| Group interviews of students. | Dec. 2016 | Video conf. | KTH members and UDSM students |
| Evaluating and planning. | June 2017 | E-mail | MIC project team (KTH, UDSM) |
| Reflective questionnaire. | July–Aug. 2017 | online | Teachers (T1–T4), Students (S1–S15), Tanesco staff (C1–C8) |
| Workshop week: Sharing the work to new CDE partners and receiving feedback. | Aug. 2017 | KTH | KTH, UDSM, DIT, and KTH GDH partners |
| Final presentations Follow-up and co-creative planning workshop for phase 2. | Jan. 2018 | DeS | MIC project team (KTH, UDSM), students, teachers, TANESCO |
| Dissemination of the lessons learned from phase 1. | April 2018 | IEEE EDUCON | PAR researchers of the project team |
| Dissemination of the lessons learned from phase 1. | July 2018 | CDIO 2018 | PAR researchers of the project team |

**Table 2.** Overview of the participatory activities that were conducted in phase 2.

| What | When | Where | Who |
|---|---|---|---|
| Planning workshop. | Jan. 2018 | CoICT (UDSM) | Project team (KTH, UDSM, DIT), teachers |
| Student recruitment. | Jun. 2018 | DIT | Project team (UDSM, DIT), teachers |
|  | Oct. 2018 | CoET (UDSM) | Project team (UDSM), teachers |
|  | Oct. 2018 | KTH | Project team (KTH, UDSM), teachers |
| Challenge definition day. | June 2018 | CoICT | UDSM & KTH teachers, DAWASA partners & stakeholders. GDH partners |
| Practical training course. | July 2018 | CoICT/DAWASA | DAWASA partners, UDSM students, DIT students |
| Final year project course. | Nov. 2018 | CoICT | Project team (KTH, UDSM, DIT), teachers, students, DAWASA |
| KTH students' arrival. | Jan. 2019 | CoICT | KTH students |
| Mid-term presentations at Research exhibition week. | March 2019 | CoICT | Project team (UDSM), students |
| Research exhibition week. | Apr. 2019 | UDSM | Project team (UDSM), students |
| 2019 IFIP WG9.4 Conference exhibitions. | Apr. 2019 | RAMADA (DSM) | Project team (UDSM), students |
| Building the demo site. | May 2019 | CoICT | Project team (UDSM) |
| Final presentations and demos. | Jun. 2019 | CoICT | Project team (KTH, UDSM, DIT), teachers DAWASA, GDH partners, other stakeholders (BORDA) |
| Presentation to Challenge owners (DAWASA). | Jun. 2019 | CoICT | Project team (UDSM), students, teachers DAWASA |
| Dar es Salaam International Trade Fair (Saba Saba). | Jul. 2019 | DSM | Project team (UDSM), students |
| Dissemination of the lessons learned from phase 1+2. | Aug. 2019 | KTH | Project team, KTH and UDSM teachers and students; GDH Teachers T21–T27 |
| Optional online reflections. | Aug. 2019 | KTH | |
| Evaluation and planning workshop. | Aug. 2019 | KTH | Project team (KTH, UDSM) and GDH |

*4.1. Participatory Action Process, Phase 1, 2016–2018*

This section will explain and share experiences from the participatory activities in the first phase of the implementation process, see Table 1. Details regarding the project topics as well as the results from the students' work are left out from the study, with a focus on the participants' experiences of the implementation process of challenge-driven education.

4.1.1. Preliminary Decisions

Early 2016, discussions were initiated between UDSM and KTH. As the vision was to have cross-national collaboration on CDE, the partners discussed how to take the first steps towards this common goal. At KTH there already existed a couple of CDE opportunities that could be used for the collaboration by admitting students from UDSM, while this was not yet the case at UDSM. Also, among the KTH team members there was an experience of participatory design research that was decided to be used in the project. Therefore the focus was set to develop a CDE course at UDSM. Opportunities were detected in two postgraduate courses that were to be developed as parts of the curriculum of computer and IT systems engineering on PhD and MSc levels at the College of ICT (CoICT) at UDSM. Colleagues from the Dar Es Salam Institute of Technology (DIT) were also involved since the idea from the beginning was to establish the joint collaboration with DIT included. This network could not be established during the planning phases and therefore this study is focusing on the implementation of CDE at UDSM only.

4.1.2. Planning Workshops in September—October 2016: The Challenge and the Course Design

The workshop for planning the course was held in September, six weeks before the actual course would start, involving all CDE implementation team members from KTH and UDSM. The aim was set that on the day the course starts, the students should face a real-life challenge where external challenge providers were involved. With this aim in mind, a road map was designed with backwards design, leading to the decision to contact staff members at TANESCO (The Tanzania Electric Supply Company Limited), an electrical supply utility company owned by the government, monitored by the Ministry of Energy (MoE), and the regulator EWURA (Energy and Water Regulatory Authority) in Tanzania. TANESCO and CoICT at UDSM already had established relations in research and development projects, while having no experience of collaborating in relation to the challenge driven education approach.

Three representatives from TANESCO were present at the following challenge-identification workshop together with the CDE implementation team members from UDSM. The KTH representatives were not participating. According to the PAR methodology this was an opportunity to make sure autonomy of the development of CDE at UDSM was established among the UDSM team members [56]. The project leader at UDSM discussed online before the challenge definition day with the colleagues at KTH. The challenge which was identified through participatory discussions, after a presentation of the challenge providers' view on current critical issues, was defined as: Inefficient processes of faults detection, identification and localization of electric supply in Tanzania.

The challenge was evaluated by the full CDE implementation team to be highly open-ended and complex. Therefore it would be possible to open up for multiple sub-challenges. On the CDE course introduction with the students the challenge and the sketch of the roadmap was introduced. Discussions were made on the demand to use systems perspectives while working on solutions, in order to understand the society, the people, the workers at TANESCO, the MoE and EWURA planning and so forth. Furthermore there would not be one single reason for the inefficiency, and there could be multiple solutions that could be argued to be of value. This was argued to enhance the training of normative competences [9].

### 4.1.3. Co-Creative Planning Workshop, December 2016

In December 2016, the project team members from UDSM and KTH, met for a 3-day co-creative workshop in Stockholm. Discussions and experience exchange were targeting current needs, namely the challenge provider relations as well as support and supervision to the students. Visits were made to locations at KTH that conducted CDE projects. A plan was made to organize a mid-term presentation day in February 2017 at UDSM, inviting all involved participants in the TANESCO challenge. Furthermore it was decided to carry out a focus group interview with the participating 15 students, in order to evaluate and plan for the coming steps so that the development would be grounded in the mutual experiences among all the students [47].

### 4.1.4. Formative Focus Group Interview with Students, December 2016

The interview with the students taking the pilot CDE project course was decided to be carried out through video link between UDSM students and two of the KTH project participants. The UDSM teachers showed awareness of power relations [61] and decided to not be involved in order for the students to speak more freely about their experiences so far.

The KTH PAR researchers asked the students one at a time to give their general reflections. This proved to be very fruitful and a feeling of trust and relatedness was perceived by the PAR researcher to emerge, after some introductory connection difficulties. Overall the students explained that they were happy to be involved in the CDE projects. When all students had shared their perspective, the researchers suggested a summary of their ideas, which was discussed and agreed upon. Usefulness, motivation and value were seen with both the content and the design of CDE. They felt high commitment and support from the teachers, who even provided with supervision on Saturdays. The critical aspects which the students and the researchers identified during this meeting were curriculum design aspects, such as workload, expected learning outcomes and assessment. Furthermore it was concluded that the relationship with the external stakeholders was difficult to establish, where the students perceived them as uninterested or unaware of CDE and quite difficult to reach. These aspects were confirmed by the UDSM teachers and an agenda for working with the identified critical issues was set for the next co-creative workshop that was planned to be held in February.

### 4.1.5. Mid-term Presentations, February 2017

All involved participants in the CDE course including staff members from TANESCO attended the mid-term presentations in February 2017. The number of TANESCO participants increased from 3 (in the previous meeting) to 12, which showed a growing collectivity [61]. The students presented their findings from their background investigations of electrical supply in Tanzania.

They described the use of traditional power system management systems which causes inefficiency with frequent power disruptions. Furthermore, they explained that maintenance response time is critically low, and the detection of faults was argued to be time consuming, which leads to higher prices of electricity. Having this background, a collective decision was made to divide the challenge into two sub-challenges, one for the Master and one for the PhD students respectively.

The MSc challenge dealt with "Monitoring and Controlling Home Appliances to Reduce Non-Optimal Power Consumption in Tanzania", while the PhD challenge addressed "Inefficient Power System Fault Prevention and Clearance". After this mid-term meeting the students started working more closely with TANESCO and the students' work became more practically oriented.

### 4.1.6. Evaluation and Co-Creative Planning, February 2017

After the more formal presentations, the mid-term meeting focused on evaluation and co-creation of the coming steps. Results from the focus-group interview were shared and workshops were organized around the critical aspects which had been revealed. Around 30 people attended, working in

mixed teams (students, teachers/supervisors and project team members from KTH and UDSM). Also peers from the Dar Es Salaam Institute of Technology (DIT) joined the workshop to bring in more voices for feedback and to nurture possible future collaborations. The groups presented their conclusions and ideas. Based on this, the project team collectively sketched the coming steps [46] and decided that the students' workload needed to be reduced, the contacts with TANESCO should be supported and increased and the UDSM teachers should work on finding links between the CDE course and all other ongoing courses in the MSc and PhD curriculum respectively.

### 4.1.7. Formative Focus Group Interview with Students, May 2017

The previous experiences from the group interview format were taken into account in order to act upon the reflections and develop the methods continuously [6]. A more informal start of the meeting made it possible to make sure everybody got a chance to enter the meeting. This time the KTH participants were aware of the tight schedule the students were having, as well as the cultural differences between the countries when it comes to starting up meetings. Furthermore, the importance of 'starting from the heart and not the head' was experienced to be given more value in Tanzania, and this was something that the Swedish project participants learned from and appreciated.

After the reflections among the participants of the interview, the students and the researchers agreed on the following summary. The students' experiences of the CDE course had evolved in a positive way since the last interview. The relations with the challenge providers had grown much tighter. The projects had become realistic and practical, and an expected project outcome was in sight. At the same time, there was still a perceived lack of prescriptions of the academic expectations on the work and the members of the CDE implementation team realized the importance of the creation of new assessment criteria.

### 4.1.8. Evaluation and Planning, June–August 2017

In June 2017 the CDE implementation project team discussed the outcomes from the group interview via e-mail. In order to create contextually relevant policy documents which would support the students' need for clear academic expectations [47], one of the professors teaching the CDE course at UDSM designed a draft proposal for the assessment criteria of the CDE courses, including how technical, team work and presentation parts should be assessed. After being ventilated and discussed, a final version was established. Via online discussions the project team also designed online reflective questionnaires for the teachers, students and challenge providers. The aim with the questionnaire was to open up for summative, deeper, individual and anonymous reflections on the quality aspects of the process [6]. In August 2017, the project team had a meeting in conjunction with the first workshop held in order to start up the KTH Global Development Hub. Lessons learned from the first phase pilot course were shared to all participants. The assessment criteria for the CDE courses were decided upon and preliminary results from the questionnaires were presented. The final reflections among the students, teachers and challenge providers in the reflective questionnaire are described in the findings section.

### 4.2. Participatory Action Process, Phase 2, 2018–2019

In this section the CDE implementation processes of the second phase will be explained, see Table 2. While the first phase was collaboratively monitored by KTH and UDSM, the leadership and autonomy in the second phase was managed contextually close by the UDSM CDE implementation team, which will be explained in the following sections. Details on the project topics and the technical work are left out, and the target is to focus on the participatory processes for change of engineering education.

### 4.2.1. Lessons Learned from Phase 1 to Apply in Phase 2

In January 2018 the project team gathered at UDSM to finalize the first phase and initiate the second. The overall positive experiences from the first phase led to the decision to focus the second

phase on developing a CDE course that could involve not only UDSM students but also KTH students on exchange at UDSM since the aim with PAR is to progress towards the shared goal [47]. The UDSM students had already started to join CDE courses at KTH and this second phase would finalize the implementation of the approach for mutual capacity building with two way student exchange and joint CDE experiences. In order to succeed, the barriers discovered in phase 1 should be targeted early in phase 2. It was concluded that the challenges the students should work with should be broader, relate more clearly to the Sustainable Development Goals in UN's 2030 Agenda and invite not only IT systems engineers but open up for collaboration among several disciplines. The relations with the challenge providers should be established earlier with a smoother start for the practical and inquiry-based steps. As with phase 1, connections should be made to external stakeholders with which the CDE course providers already had good connections. The schedule and overall curriculum should be better planned so that teachers and students would have more appropriate time to work on the projects. All involved should be introduced more deeply to concepts and processes related to CDE, through shared experiences among all stakeholders from the first phase [61].

Aspects concerning the international exchange were raised which had not been tested in the first phase of the project. Formal and administrative routines between the International offices at each university which is key for any exchange to happen were established.

The next step would be to discuss where in the curriculum UDSM could organize a CDE activity for both UDSM students and students from KTH and from other disciplines and even other universities.

### 4.2.2. Initial Decisions for CDE in Phase 2 at UDSM

The lessons learned from phase 1 motivated the use of CDE in undergraduate teaching at UDSM. The existing *Final Year Project* (FYP), running for two semesters from November–July, was seen as a suitable course to choose, due to its flexibility of contents. The choice of a BSc level for the CDE implementation opened up the possibilities for KTH BSc thesis students to apply for doing their project at UDSM during their spring term from January–June and become part of the second half of the CDE projects. There were concerns though from both sides whether this would be feasible, in terms of logistics, interest, knowledge level and less number of weeks to work on the challenges as compared with the phase 1 course on MSc and PhD level.

The existing team of teachers connected to the FYP had the possibilities to influence the parallel courses which the majority of the students would attend. Furthermore the FYP coordinators had been involved in the first phase of the CDE implementation as supervisors and members of the CDE implementation team, with gained knowledge and experience. Therefor the time was perceived to be right to place the full leadership and autonomy of the second phase at UDSM, since the aim with PAR is to gradually make sure the ownership is in the hands of them who are to work with the new [56]. It was decided that the KTH team would provide with support when asked for, as well as organizing the final evaluations. The first decision made by the UDSM team was to try to take advantage of the mandatory 8 weeks of *Practical Training* (PT) that their students have during July–September. During the PT, the students are exploring practical applications of their studies outside the university.

Due to the challenges with water in the community, the idea to work with the water sector was elaborated. The management of the water sector was contacted and introduced to the project which supported the idea. The CDE coordinating team therefor contacted the Dar Es Salaam Water and Sewerage Authority (DAWASA) which had good relations with UDSM since earlier. The DAWASA team agreed to contribute with challenges and supervision, after the work from phase 1 with TANESCO had been explained. DAWASA also agreed to invite the students to their sites and provide with feedback and support during the PT.

### 4.2.3. Planning, Challenge Definition and Recruitment

In May 2018, a challenge definition day was organized at UDSM. This was a co-creative workshop carried out by teachers in the field of electrical engineering as well as water management from UDSM,

DIT and KTH, engineers from DAWASA, representatives from the KTH Global Development Hub, as well as other stakeholders in the water sector. By this, lessons learned from phase 1 on a confusing start, as well as lack of tools for challenge analysis, was used to develop the processes. The overarching challenge was the water and sanitation situation in the Dar Es Salaam area, and the challenges which were analyzed to be open-ended and complex enough to open up for various interpretations were formulated as:

- Lack of Infrastructure and Asset Management;
- Inefficient Customer Care, Billing and Revenue Collection;
- Low Sanitation Coverage;
- Access to Good Quality Water in Unplanned Areas.

A follow-up workshop involving 14 teachers from UDSM, DIT and KTH who would supervise the students, was initiated. The idea was to set forth the development of more specified challenge statements for each of the four identified challenges, as well as plans for the student projects. This was decided in order to act upon the experiences shared by the students from phase 1, on the unfamiliarity with the challenge statements as well as the heavy workload they encountered in their studies. One critical issue that was raised during the workshop with the supervisors from UDSM, DIT and KTH, was the possible impact on student team relations due to the differing academic calendars. In particular, it was noted that KTH students would only be able to join the project teams in Tanzania two months after the students in Tanzania had begun their final year project. It was decided that the teachers from each of the institutions would strive to minimize the effects of this difference, by supporting the students in their respective institutions. Additionally, it was agreed that online meetings would be organized for the KTH students prior to their arrival. Other key items were also raised and planned for, including training workshops for teachers and stakeholders on the concept and practical application of CDE, in order to use the accumulated and growing knowledge and experience among the teachers at UDSM [47].

Due to the broad nature of the challenges this opened up for a recruitment of bachelor thesis and final year project students in several fields, which was also an important shared aim with the CDE implementation. The recruitment at KTH, DIT and UDSM ended up with 39 students in the field of electronics and telecommunications engineering, computer science and engineering, civil engineering, management and engineering, energy and environment and electrical engineering.

4.2.4. Practical Training, July–September 2018

In July 2018, methodologies for working with design and solutions on societal challenges were introduced to the students and the teachers in the same time period as the challenges were introduced. This was a step to further the training of appropriate methods for understanding the challenge, for ideation of solutions towards different criteria, as well as methods for implementation and follow-up. The students' task was to suggest sub-challenges related to the four main challenges from DAWASA (see 4.2.3). They were not introduced to the solutions from the supervisors' workshop in May 2018, in order to establish autonomy among the students [56]. Visits and interviews at DAWASA as well as with customers (citizens) informed their work on a root-cause analysis. Their result was almost the same as the teachers' suggestion for sub-challenges which was communicated to the students in order to give them early experience of success.

In August 2018 all students, teachers at UDSM as well as the external challenge providers gathered for a two-day workshop. The students shared their analysis and understanding of the challenges following their application of the design thinking methodology. A feedback session was organized where the teachers and the challenge providers further shaped the students' work, preparing them for a new study tour of DAWASA's infrastructure and subsequent attachment at DAWASA which followed. At the workshop, deeper discussions were organized on how the challenges were related to the SDGs in UN's 2030 Agenda [1], as well as discussions about CDE in general.

By the end of the eight weeks of the practical training, the students had formed teams of up to 4 students and developed roughly 16 sub-challenges which were further broken down into individual projects for each student within each team to carry out as their final year project. The developed sub-challenges were shared with the teachers from KTH who were to carry out recruitment of their students the following month.

### 4.2.5. Final Year Project, November 2018–July 2019

Students from UDSM and KTH joined the CDE activities at different times during the final year project (FYP) course. On both occasions, the students were oriented with CDE and the teams' tasks were reformulated in order to accommodate the incoming students as smoothly as possible. The incoming KTH students, together with KTH teachers, were also given a tour of DAWASA's infrastructure to familiarize them with the situation on the ground for their understanding of the challenges.

In January 2019, following the addition of the remaining students, a workshop was conducted that focused on the development of work plans. The teachers assessed the students to be confident in the disciplinary aspects of their work, as well as in their work at the sub-challenge level. It was however evident that students' teamwork should be improved at the challenge and inter-challenge levels. Despite the situation where all students were finally being physically present, differences in the students' timetables and the physical distances between the students' colleges limited the group sessions that were planned, calling for some to be shifted to weekends.

A meeting with all supervisors was initiated by the CDE implementation team at UDSM, to provide an arena for the supervisors to share the difficulties they encounter [47]. It was collectively decided to follow up with a workshop on team supervision for teachers from UDSM, DIT and KTH [46]. The workshop was open for more teachers at UDSM, in line with creating a collective growth for durable change [61]. 27 participants joined the workshop which was found to be very useful in arriving at methodologies for the successful supervision of student teams towards intended learning objectives.

The first presentations from the fully formed teams were given to stakeholders at the end of January 2019. The teams presented initial designs of proposed solutions for feedback, before delving into the development of prototypes for their proposed solutions. After implementation and evaluation, several of the teams reached to working prototypes that were ready for deployment. A demo site was constructed at UDSM to mimic DAWASA's water distribution network. The students tested their prototypes on the demo site which was used to give the final presentations of the solutions to stakeholders in June 2019.

### 4.2.6. Final reflective seminar, August 2019

In August 2019 the CDE implementation team, the teachers and student representatives, who had been involved in the first and/or the second phase gathered at KTH. All groups were asked to prepare their reflections and share this orally during a one week reflection and workshop seminar. Also, an online questionnaire was available for all who wanted to add their reflections there. The presentations were recorded, transcribed and thematically summarized in Section 5, along with the findings from the first phase.

## 5. The Students', Teachers' and Challenge Providers' Perceptions of the Implementation of CDE

The findings on the perceptions described by the students, the teachers as well as the challenge providers, of the implementation of challenge-driven education, in the final reflective questionnaires and the final reflective seminars after the first and the second phase, are presented in the following sub-sections. When quotes are presented, they come from the reflective questionnaires, and the responders are noted as the following. From the first phase, the students are S1–S15; the teachers are T1–T4 and the challenge providers are C1–C8. From the second phase, where much more emphasis was on the oral reflections, there were still some who used the online questionnaire, and the teachers

from KTH are noted as $T_21$–$T_24$, while the teachers from UDSM are $T_25$–$T_27$. The only student who used the online questionnaire will not be quoted but his/her thoughts are embedded in the summary.

*5.1. The Students' Experiences of the CDE Implementation*

For the UDSM students in the first phase, practically oriented project-based courses and working with the involvement of external stakeholders, has been a completely new experience. The approach of working in a project based setting has been demanding and yet very rewarding according to the students in both phases. The "transition to challenge based was not so smooth as we are used to study for exams and not for solution provision and working with people from industry", argues S14, who believes this demands a "new approach especially from the supervisors' mindset". There is a clear perceived contrast between this setting and the previous learning experiences from "traditional education where we would be just focusing on passing the course", argued by S14. The students experience that they start to expand their ways to find knowledge in "various literature", according to S2. S7 writes how this new approach "gave students the chance to consult books, papers and other learning materials by themselves". The students in the second phase, who were on BSc level, realized that while not having so much knowledge in the field, they could also learn and develop as they worked with the challenge. The second phase opened up for clearer comparisons between the two contexts since KTH and UDSM students met and worked on projects together. Something that really stood out according to the KTH students was that the CDE approach was very focused on finding solutions. Ordinary BSc theses at KTH are more theoretical and not actually aimed at designing products, according to the students from KTH.

Team-based learning is an integral part of the CDE course setting in this study, which for many of the students from both phases of the implementation seem to be quite a radical difference from how they are used to study. Suddenly, their peers' knowledge and skills have an importance for the accomplishment of the task. Furthermore, how each individual works, studies and contributes, become important pre-conditions for not only themselves, but in a much larger context. The team based setting, according to S1, "made students to struggle to not be seen by others as a burden towards delivering of the project". S1 states that "if other fails to deliver the whole group has failed". S4 pictures the team as a "catalyst towards working independently as always one must have something to contribute to the group". This contribution is emphasized also by S8 who states that "most of our individual tasks depend on one another", and by S13 who argues that they all collaborate to "make the entire system to work ( … ) to accomplish a common goal". In the second phase the team-based learning was more emphasized in the introduction to CDE and how supervisors were trained. This was noticed in the reflections made by the students in the final reflection week. They argued that the way of finding solutions mainly comes through collaborative approaches on interpreting problems and understanding challenge providers. Furthermore, in phase 2, the students collaborated over new borders. This is noticed in the reflections, where students in the second phase who were from UDSM where the CDE course was given, pointed out that students from other countries, universities and disciplines should be better integrated and involved even if starting later.

A high motivation is expressed by all students involved in both phases. The students describe usefulness, meaning and value with the format, contents and ideas of the CDE course. S8 writes that "the project is a real life challenge in Tanzania and many developing countries and I feel happy and grateful to get an opportunity to work with this project in an academic context". They also emphasize the importance of being part of making this reality better, "knowing that I am working on something practical which is going to solve real world problems", as S9 argues. In this way they felt real usefulness of their own skills and knowledge, as this could be applied in the work like for example "knowledge in embedded systems, programming and designing and databases", according to S1. Overall the CDE has been a very rewarding experience, where gaining confidence of being able to contribute to real challenges stands out as a main strength, as expressed on the final seminar of the full project. The students in phase 2 from UDSM argue in the final seminar that they hope that the CDE initiative is

the start of something that will be integrated at the full university and that it will reach more students as well as more people outside the university. The students from KTH too from their side find that taking a CDE semester at UDSM is highly recommended.

The teachers' role has been crucial for the students in both phases. The students raise that they felt well supported by their teachers already in the first group interview of phase 1. S6 sees the teachers as team members, and S5 realizes that with the fact that the teachers are no longer "feeder of materials", students become more creative. S10 agrees and finds that while previously waiting for the teachers to make the planning, he/she thinks and acts more early on his own. S1 compares with traditional education and argues that "lecturing doesn't put students close enough with teachers, so even the sense of belonging is not there". In this setting, where also challenge providers act as supervisors and experts, is seen as "the perfect knowledge combo", according to S6. The training of the teachers' skills on designing CDE activities for the students seemed to have been fruitful. The students lifted in the final seminar that they thought the training and activities the teachers carried out were inspiring, such as a pitch training session. The KTH students in phase 2 expressed that they felt warmly welcomed by the teachers, and were happy to be part of something new. They also expressed in their oral feedback at the seminar that they were aware that they were pilot students and took for granted that there would be critical issues to solve. Since they arrived late, and there had been trouble with the preparatory online meetings, they had preferred deeper descriptions from the teachers of the challenges they would be working with, before they arrived in Tanzania. Another key issue was the unclearness whether the KTH students had a specific supervisor at UDSM or not. They did have academic supervisors at KTH, but would have wanted to understand better if there was a specific teacher at UDSM or not that they knew they could contact directly. In the KTH system, on the final thesis, there is normally a specific teacher and not a team of teachers who is your supervisor.

Collaborating with challenge providers from outside the university has been perceived as very rewarding but also very new and challenging from the students' side in both phases. Some of the students in the first phase felt that the challenge providers were a bit reluctant in the beginning. As the work proceeded, they realized that the reason was probably that the challenge providers were "not aware about the approach", as S8 writes. S6 emphasizes that "in the end you can tell the huge difference ( . . . )" when "stakeholders were very cooperative and their input was very significant". In the second phase, where the students used their practical training for becoming familiar with the context for the challenge owners, the mutual understanding was much better established as reflected on the final seminar where students from both phases were represented. S1 in the first phase argues that the stakeholders' "appreciations, comments and recommendation built a hard working spirit and a feeling of not letting the university down, nor the supervisors or ourselves". S7 claims that the challenge providers are the ones who "bridge the gap between industry and academy". The challenge providers help the students understanding the problem better, as well as how to frame and understand the context of the problem. In the students' reflective questionnaires after the first phase, we see that they mention the system perspective and the holistic approach to a problem. S7 writes that this perspective "has introduced me to the idea that, when solving a particular problem, I have to consider how it will integrate and co-exist with available or upcoming solutions". S7 argues that it is crucial to understand the user and the site requirements: "At the beginning we had our opinions of the problems facing the energy industry, particularly the main electrical company. However, when we met them, they had most of our listed problems solved under various stages of implementation. The lesson learnt was that, we should have started on their side". In the second phase the integration of the Practical Training course in the CDE implementation was found very valuable by the UDSM students in the final reflection seminar. Opportunities to create good relations with the challenge providers and to understand work methods and tools were appreciated.

The challenge providers, according to the students, help with understanding different solution processes that are common out in the field of engineering practice. S11 finds this to be "very helpful since the stakeholders were taking us through the working principles of the systems,

rectifying, criticizing and suggesting the best we could do to add value to our solutions". Students from both phases experienced a common frustration when the scope of the project changed with every interaction with the challenge provider. The students from UDSM in phase 2 suggest in the final reflection seminar, that a space could be created, where people from the society are invited to work on and discuss the challenges more continuously. The contact person at DAWASA was very helpful according to the students from UDSM and KTH, and he seemed very committed in his work to bridge the contacts. The KTH students argued that he was heavily overloaded with supporting the contacts for the 39 students.

Curriculum design, students' workload and previous knowledge are impending factors in the light of the new and unknown ways of working in the CDE setting, compared with the traditional education. The overall workload in the program, including all parallel courses, is the main critical aspect that is raised by the students in the first phase of the implementation. This was specially noted in the mid-term evaluation. The curriculum design for the students in the second phase was better planned with regards to the overall workload. Still, they expressed in the final seminar week, a feeling of limited amount of time, where the reason was mainly on their perceived need for more background knowledge and skills. The second phase students at UDSM recommend that the projects should start earlier by having the challenges defined earlier, despite the significant development of this time for preparation between the phases. In the second phase even more issues concerning curriculum design, workload and previous knowledge was opened for due to the international collaboration. Due to the different timings and organizations of the academic year at the two universities, the students at KTH entered the projects in Tanzania when the UDSM students were already well familiar with the challenge and had been working with the project for a couple of months. The preparatory online meeting that was held, was short-lived due to Internet connectivity problems, which made it difficult for the KTH students to prepare.

The assessment of the students' learning and project work has been a crucial aspect for the students in the first phase. In the group interviews, the students raised the issue that the faculty's as well as the challenge providers' assessment of the performance of the students' work needed to be explained and discussed. The students wondered how these new activities in their academic studies would be graded. All of the students and teachers had mainly previous experience of individually performed exams. The perception among some of the students was also that the expectations from the challenge providers' side were a bit too high. S1 writes: "I remember some stakeholders thought this is a research work of our dissertations, but teachers and we students were clarifying in our presentations the mode of program as just one of the work and not a research work". Some improvements on the perceptions of the assessment could be noticed in phase 2. Here instead new issues arise, where there was a sense among the KTH students of not knowing if the academic work at UDSM would suffice when coming back to KTH for assessment of their final BSc thesis, as raised during the final reflection seminar.

*5.2. The Teachers' Experiences of the CDE Implementation*

The teachers perceive the influence of CDE on the students' development in both phases of the implementation process to be significantly motivating. This happens since "the possibility of the activities ending up being too academic becomes nullified", according to T1. T4 argues that "the course supported students' ability to work independently by getting a rare opportunity of dealing with challenges from outside the campus. Students get the opportunity to mingle with stakeholders straight away". Also the teachers think that the students seem to be challenged in a constructive way. T3 tells that "the students find it challenging to visualize realistic challenges as an academic problem because they are used to seeing on problem being done by an individual but a real life challenge does not generate a single academic problem". The integration of the Practical Training course in the second phase of the CDE implementation has according to the teachers in the final reflection seminar provided an invaluable learning environment for the students.

The new situation where the students are studying and learning quite differently compared with traditional education activities is often mentioned by the teachers. While the students are becoming more independent in relation to the teachers and the normal ways to go about learning, they are also developing and improving their relations to their peers, as team members with a shared goal. T4 explains that this becomes visible since "each group must know the knowledge, skills and experiences of every group member" for success. The overall workload though during phase 1 is considered to be too high. The schedule for the students will need to be tuned. One way of dealing with this issue can be to "conduct the other courses with basis on the challenges in hand", writes T1. In this way the CDE course can be better integrated in the study program. One of the CDE project team members at UDSM, who has been supervising students in both phase 1 and 2, explained on the final reflection seminar that he was surprised to see that CDE was possible on the bachelor level, where the numbers of hours to work on the project is far less than for the master's and PhD students. During phase two a strong teacher team with experience from the first phase locally at UDSM made this possible. A supervisor from KTH admits during the final seminar of phase 2 that he was wondering how the BSc students would cope with the challenges and was happily surprised to see their progress through the work. The teachers agree on the final seminar that the students are walking the extra mile in the CDE setting, compared to the ordinary routines when they are studying for an exam. With CDE, a KTH supervisor concludes, the students create their own capacity.

The students' learning outcomes that the teachers from UDSM and KTH highlight after two phases of CDE implementation are an increased confidence, an awareness of the contextual aspects and being better equipped with processes for continuous learning. The teachers argue that they find that they can be proud of the achievements made, and also that the students are reaching higher levels of learning than compared with traditional courses. CDE leads to "heavy engagement" from both students and teachers, as the KTH teacher $T_2 1$ phrases it. For instance, it is brought up on the final seminar, the CDE students have been approved to present their work on the research exhibition week to a much larger extent than any other course groups at UDSM. Still, some teachers think the level of achievement can be improved even further, aiming at more real impact in society. In order to promote and deepen the students' work with sustainable development in the projects, $T_2 4$ a supervisor in the second phase from KTH argues in the online questionnaire that students will need to reflect on sustainability through the CDE projects and teachers should challenge this even further to promote students to develop their knowledge in sustainability issues. $T_2 2$ argues that the CDE setting makes this possible, the projects are "eye-opening". $T_2 6$ at UDSM brings up improvement ideas after the second phase on how students should be better prepared for the CDE course, where a much earlier start should make sure the projects can be accomplished. Furthermore he/she thinks that "more time should be given to students to finish their projects even after they have graduated. This will provide enough time for solutions to get customers".

The broader intake of students from different disciplines, from phase 1 to phase 2, is also effecting the discussions about the students. The UDSM teacher $T_2 7$ raises the benefits with the multidisciplinary approach so that students become better at collaboration, which opens up for better capacity to solve challenges and it also improves the employability of the students. $T_2 5$, at UDSM, though thinks that the interdisciplinary approaches should be strengthened since the set-up today is more multidisciplinary. The supervisors at UDSM as well as KTH argue in the final seminar that one should make more use of the several disciplines involved as well as the international setting. Some students are good in the technical laboratories, while some are good out in the field among citizens and other students are better in the field of finance and economy. This should be supported better in the projects. The CDE organizers should discuss more what the learning scope should be since students have mixed backgrounds, and design activities that make more use of this, instead of leaving it all to the students. This would also contribute to more collaboration among students from different countries, universities and disciplines. Furthermore the argument is made by the CDE implementation leaders at UDSM, after the second phase that a crucial step will be to evaluate the assessment and evaluation methodology of students'

learning at UDSM and KTH, since this is not always compatible with the type of knowledge and skills they want the students to develop.

Regarding the role as a teacher, in phase 1 it was the first time experience, for most of the teachers, of teaching in a project based course setting. Since the students are working quite differently in a CDE course, the teachers notice that their own mindset is changing as well "because the students have from the beginning known that they own the challenge", as T1 writes. As well, T2 pictures the teachers' role to be to "democratically allowing students to identify their challenges, formulate method and solutions". In order to infuse courage, creativity and innovation skills among students, T3 argues for the importance of the teachers to not "dictating exactly and how to go about in their work". One of the teachers, T3, tells that "we had to use weekends for supervision not to interfere with their students' regular activities". It becomes rewarding to work as teachers in this CDE setting, argues T2, "by seeing the real and immediate impact". After the second phase, the teachers at UDSM who have been involved in the full implementation process express in the final reflection seminar, the clear difference between the phases regarding the level of stress for the teachers. In the second phase there was an in-house experience of being a teacher in a CDE setting, and the teachers knew they could coach and advice each other.

Though the UDSM teachers understand that they have come far, and after the implementation project now have several champion CDE teachers, the CDE implementation leaders at UDSM argue that maintaining this approach will not be easy. They express that they are in a process of continuous learning what CDE is and how to work in that format. This they argue will demand lots of commitment. The new teachers who entered the implementation process during the second phase argue on the reflective seminar though that the barrier is not a lack of commitment among the teachers at UDSM. Rather, the main barrier is the unfamiliarity with the CDE methodology. He explains that when teachers are called in to be supervisors, they are not familiar with the concepts and methods of CDE. The ordinary work relates more to hard engineering, and the mode of delivery is very different from this setting. The implementation of cross-border education in phase 2 also implies that teachers and supervisors will need to work across borders. A supervisor at KTH argues that teachers and supervisors from all disciplines and universities involved in the projects are key to a future success and the barriers for this need to be removed. In order to keep collaborating among the universities, $T_2 2$ from KTH thinks the logistics should be improved. $T_2 3$ finds that the discussions around CDE becomes too theoretical, and the "focus should be more on actions"—"we need more practice". $T_2 7$ at UDSM, writes in the online evaluation on four improvement areas regarding this. A well-documented guide on how to conduct CDE should be developed. The assessment of students on individual and team-based level still needs clarification. The teachers should be well-trained on how to supervise students in teams. Finally this teacher argues that financial support is needed in the CDE projects, to cover the costs of prototyping tools.

To collaborate with external stakeholders as challenge providers, and inviting them to be advising on and monitoring the course topics, has never been tried out by the UDSM teachers before phase 1. T4 has earlier found the boundaries between the university and the society to be very limiting while at the same time "globalization effects are felt daily". The teachers perceived their relations with the TANESCO staff members to be a bit shallow in the first parts of the CDE course. T3 explains that it was as if the challenge providers didn't seem to be curious or interested. However, once they had started working on the challenge, the relations evolved and "at the end of the course they the challenge providers expressed interest to involve the College whenever they will need to evaluate technology related issues". T1 argues that the continuous meetings between students and stakeholders provides with frequent feedback which "managed to re-align the students to the real challenge each time there is a meeting so that the students do not come up with unrealistic, impractical solutions".

By having challenge providers involved, also the teachers find that they themselves can provide with better supervision: "The stakeholders' inputs help to guide the supervision work so that the students work on what is achievable", according to T1. "CDE makes academia win trust from challenge

owners/society", which will enable further collaboration is argued by $T_2$7 at UDSM, after the second phase of the implementation. On areas to improve, the main area that the teachers bring forward after the first phase is the importance of developing tools and processes for the work with the definition and understanding of the challenges and the problem formulations together with the stakeholders. After the second phase it is noticeable how the lessons learned from phase 1 has been applied. In phase 2, the project team more clearly embedded design thinking methodologies. The UDSM teacher $T_2$7 explains that the methodology "of the way complex challenges facing the society can be solved to have an impact to the society where the challenges are derived" is perceived to be well defined.

How to build and maintain good relations with society, and to see high commitment from challenge providers during the project work, is a crucial issue to continuously monitor, which is mentioned by both students and teachers from KTH and UDSM. A clear difference between phase 1 and 2, with regards to the relations with society, has been the emphasis on the Sustainable Development Goals as a framework for the challenge definitions. After the second phase it is concluded, among the participants of the final reflection seminar that working with the Sustainable Development Goals as a starting point should be more or less evident, since almost all challenges faced in society are inter-linked with them. This should continuously be monitored though. One of the teachers at KTH, $T_2$4 argues in the final online questionnaire that his/her experience from previous work is that the challenges that are provided are not always as open as desired. $T_2$4 suggests the creation of a network for challenge providers, professors and more as a platform for these discussions. One of the UDSM teachers, $T_2$6 argues in the online survey, that deeper follow-up among users and customers, in order to report on impact, is crucial for the continuous alignment with the sustainable development goals. Furthermore, $T_2$7 argues that the "sustainability perspective must be integrated in the study curriculum so that the students are aware of what is involved in sustainable development goals even before getting to solve the real-challenges facing the community". The KTH teachers agree, and one of the teachers points out on the final seminar that to be working with the SDGs as a starting point broadens up the earlier work with water, where the previous focus mainly was on drinking water and sanitation. With an SDG perspective, the area widens to management of water, wastewater as well as ecosystem resources. Means to reach the goals and indicators should be a combination of financial, technical, capacity-building, digital and partnership instruments. Furthermore the perspective is even broader, since solving the SDG6 (on water) would imply much better results on many of the other SDGs. The conclusion is made on the final seminar that all contributions from higher education of working on solutions is key for the future, so that universities are relevant and useful for a sustainable society. In the final online questionnaire, the KTH teacher $T_2$3 argues that the benefit with CDE is "the fact that it is a very effective way to integrate sustainable development with several other general engineering skills, and that it can be implemented in courses of different type and extension".

### 5.3. The Challenge Providers' Experiences of the CDE Implementation

In the first phase the TANESCO member C2 had the impression in the start of the CDE discussions that "it sounded like an impractical imagination". C1 thinks that this vagueness continued in the early stages of the project, and the students "seemed not to follow the comments" from the challenge providers. C4 points out as well, that he/she thought the projects were quite difficult to understand. As well, C5 pictured the projects as "shallow".

The increased number of actively involved industry and government stakeholders in the CDE project phase 1 seems to go hand in hand with a continuously improved sharing of understanding among the students and the challenge providers. C2 argues that "with time and presentations encounters the feasibility unfolded". From the challenge providers' point of view, it seems as they realize, that the more the students got the chance to come out to the reality, the better they could take advantage of the feedback and information received. C6 writes: "After visiting TANESCO and they got the overview of power management system, then they came up with the real solution". C1 noticed

that during the field visits, the students are open-minded and eager to learn, and they managed to "change the scope and approach in order to solve the challenges we are facing".

After the first study year with CDE, the TANESCO staff members expressed that they were satisfied both with the outcomes of the students' projects, as well as the idea behind CDE. In the everyday work at TANESCO, there is little time for reflection and research. C1 points out that it has been quite unique to have talented minds with "time for research study". This has opened up for a free dialogue, according to C2, and a "partnership and shared understanding of the motive behind the methodologies for the program. This has also been the key to success in meeting deadlines and having a working solution".

There were noticeable differences in the perceptions among the new stakeholders in the second phase. The commitment from DAWASA was high from an early start, which the providing of the possibility to organize the 8 weeks of practical training at their site is a clear sign of. The challenge providers in phase 2 describe the water challenges facing the society as enormous and that it takes a long time to come up with solutions. Furthermore they argue that they could from the start see that the CDE approach would make their work easier. With the students' investigations, prototyping and testing at the sites, it more likely leads to locally relevant solutions. Otherwise, they argue, with little time for investigation, products are often purchased that have been developed at other places and are not customised to the setup at DAWASA. What the challenge providers can contribute with, according to the DAWASA partners, is their experience and commitment. When asked to give their final reflections on hot to involve more challenge providers in the engineering education, they argue that once a challenge provider can see the real benefit, and also understand their own role in the project, then a successful collaboration in the CDE can happen. To be convinced about the possible win-win situations is easier when you are informed by previous real-life projects and their impact.

The output of the students' work in CDE phase 1 and 2 was in several projects regarded as directly useful. Several planning projects are now ongoing to have some of the proposals implemented in real-life quite soon. Furthermore, planning is carried out to implement the coming steps in phases.

## 6. Activity Theory Analysis of the Implementation of CDE

In this section the perceptions of the implementation of CDE among the students, teachers and the challenge providers are analyzed with activity theory (AT) as the framework. The essence of the relation between the subjects (students, teachers and challenge providers) and the respective components (object, tools, community, division of labor and norms) is described in the following sub-sections, and graphically presented in Figure 3.

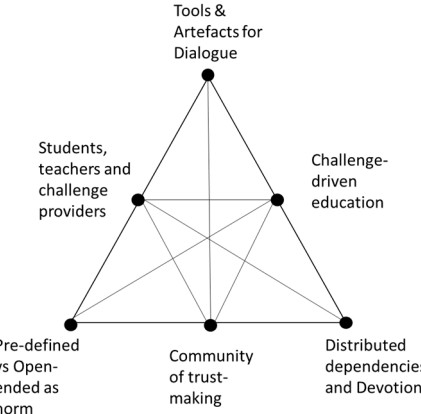

**Figure 3.** Activity theory analysis of the engineering education activity system in the implementation of CDE.

### 6.1. Subject—Object: Relevant and Rewarding

While the reluctance towards the object CDE is noticed in the first parts of the implementation, especially among the challenge providers, who even perceived it as "impractical imaginations", the essence of the perceptions among all subject groups is that CDE is relevant, useful and rewarding. The students perceive a gained confidence and meaningfulness, and their gained learning outcome is regarded by the teachers as better than in traditional teaching. The challenge providers see the rewards with the time invested on analysis, research and development related to their challenges. The teachers find it motivating to teach students who are actively involved in the learning activities, and they themselves find that they are walking the extra mile, as well as gaining from the collaboration with the external stakeholders. All in all, the subject–object relation could be referred to as the main driver for the change to happen and the implementation plans to succeed. Furthermore, both the students and the teachers express that they want the CDE projects to expand, both in terms of time to work on the solutions and the implementations, as well as to the number of students and teachers who would have the possibility to study and work in a CDE format.

### 6.2. Subject—Tools: Tools and Artefacts for Dialogue a Necessity

The written guide from KTH which was one of the first artefacts that was developed for the implementation, was early recognized as interesting but not sufficient. Through the phases of the realization of CDE at UDSM, the artefacts and tools that stand out as promising and rewarding, are all aiming at dialogue rather than a one-way communication. The study visits at the electrical supply company was the first eye-opener for the three subject groups in the community on the importance of meeting to learn from each other. Feedback as a tool for success in CDE is discovered in both phases among all subject groups. From the students' side it's told that "appreciations, comments and recommendations build a hard working spirit", while it is argued from the teachers' perspective that "the stakeholders' input help to guide the supervision work so that the students work on what is achievable" and on the challenge providers' side it is stated that this dialogue is crucial for the "success in meeting deadlines and having a working solution". Several of the improvement needs from the first to the second phase that are related to dialogue, are to establish relations with stakeholders earlier; to share the experiences among the first phase participants to the second phase as well as the need to learn about how to communicate with and understand the perspectives of different actors. The main successful actions between the two phases of the implementation project, is argued in the final reflection seminar to be the integration of practical training (PT) in the CDE, the workshops and training on design methodologies and tools for challenge analysis, as well as the coaching and supervision workshops for the teachers. All these actions could be re-labeled as tools and artefacts for dialogue. After the second phase, the artefacts which are suggested to be developed, are makers and meeting spaces where participants can meet and learn from each other, as well as a network formed by CDE teachers and challenge providers. This analysis reveals that when implementing CDE and building new communities around the solution of societal challenges, the development of tools and artefacts for dialogue, in different forms should be followed up and supported.

### 6.3. Subject—Community: Building Trust in Yourself and the Community Is Essential

Through the implementation phases there has been an increase of the complexity of the community. The students and teachers come from a diversity of backgrounds, disciplines and countries, and are working with societal stakeholders from various areas. What is prominent in the perceptions among the subjects with regards to the community, is the sense of the need for trust. The challenge providers state that they saw little reasons in the beginning to have students working on their real-life challenges, while after having giving it a try, due to previous good relations with the university in other areas, they express a clear understanding of the students' contributions. Here, testimonials of previous success between the different phases have been the key to establish this trust much sooner. The students

too are continuously reflecting on the trust in themselves, the trust in their peers "to accomplish a common goal", the teachers as well as the challenge providers. Continuous meetings with feedback on the achieved milestones have been crucial to build this trust among the students. The teachers describe that they thought the students wouldn't be able to deal with the challenges and in their reflections they share their surprise moments when the students deliver much more than expected. Furthermore trust in oneself is also an issue for the teachers who struggle with the question whether they are able to work in this new format and able to support the students in challenges and areas they know little of themselves. Through the dialogue with the challenge providers, trust among the teachers is gained on the feasibility or improvement needs of the feedback and supervision they have provided to the students. This analysis of the essence of the subject – community relations in the activity system reveals that one needs to continuously evaluate, monitor and develop the perceptions of trust among the community members.

*6.4. Subject—Division of Labor: Shifting Dependencies and Devotion*

The division of labor as a point of measuring conditions for change provides with the understanding that the roles are shifting which is also true for the distribution of power among the community member groups. In the engineering education setting where CDE is implemented, shifting dependencies can be argued to be the essence of the relation between the subject and the division of labor. Faced with the real-life and open-ended challenges, the teachers are not in the position to dictate the students "exactly and how to go about in their work" as one of the teachers argues. Among the students, the distribution of power is revealed in their expressions of the new dependency of each other, where "most of our individual tasks depend on one another" and "if other fails to deliver the whole group has failed". The challenge providers who normally work as engineers in the water or the electrical supply area, enters a new arena with the CDE. While seeing little use of CDE in the beginning, they soon build new dependencies where the students and teachers provide them with deeper analyses on their work, and their own practical experience and feedback become valuable in new ways. With the in-house experience among the faculty as teachers in a CDE setting, the stress is reduced between the two phases of implementation, but the workload is overall very high for all subject groups. They express this by stating they will "walk the extra mile", "work on Saturdays" and "not letting the university down". This is on the one hand a sign of high motivation, but can also cause overload and stress. Therefore, when implementing CDE it will be of value to study the participants' perceived dependencies, shifting roles and power, including how well they can balance their high devotion with a feasible workload.

*6.5. Subject—Norms and Rules: The Prepared vs the Open-ended Creates a Negotiation on What Education Really Is*

In the analysis of the relation between the subject and the norms and rules in the system, the contradictions or conflicting pre-assumptions and surprises seem to be related to the clashes between the open-ended approach and the traditional education format. In the traditional format, the teachers and the university system have material, outcomes and assessment organized in advance of the actual learning activities. With CDE there is a clear shift. The teachers have no 'hidden' or correct answers and the students can no longer act and digest as previously. This leads to the development of the activity system so that new tools, new communities and a change of the division of labor are emerging, as presented in the previous sub-sections. Still, rigid organizational structures are contradicting the development. Throughout the implementation, the question about the assessment of learning is hanging over the participants. For the UDSM students, the question on how they could use this work in the final exam was impending, and from the KTH students' side there were doubts whether their project work in Dar Es Salam would count as a BSc thesis when coming back to the home university. Another clear contradiction for the students in both phases and from both countries was that the schedule and timing of the project work was enormously difficult to squeeze in to the regular

pace of academic studies. The analysis reveals that when CDE is implemented in the engineering activity system the question on what engineering education really is seems to emerge, a question that should be important to attend.

## 7. Discussion

The current and future challenges in the world calls for a change of education [9], which will encounter barriers such as attitudes, competence and the organizational structures [34], in a situation where the majority of teachers who are implementing SD in their teaching are themselves in the fields of sustainability [8]. This three-year project shows that CDE can function as a motivating driving force to integrate ESD in engineering education, together with engineering teachers and students in a variety of technical fields. The high motivation among students and teachers to work in the CDE format, to walk the extra mile, to develop new patterns of actions and to reach higher levels of learning outcomes, which is crucial for lasting change [20,22,23,32,34] is evident in the data from the two phases of implementation of CDE. The motivation is also high among the challenge providers once they are on board and see the benefits for them with CDE. The intrinsically motivated transformation of the participants' roles and activities can help us understand initiators [20–22] for engineering education change for sustainable development. In the light of the high motivation among the participants in relation to CDE, one can argue that the concept of sustainable development can be attractive, feasible and mutually understood among students and teachers, with the help of CDE, the *what* of the change [3,22]. In order to reach out to more teachers and educations to integrate ESD in innovative, inspiring and useful ways, we argue that it is successful to aim for this type of practically oriented initiatives, focusing on real-life competencies, for example by working with socio-technical challenges as opposed to the focus on SD theories.

Malmqvist et al. [12] provide with an overview of CDE initiatives which reveals the risks with CDE ending up as extra-curricular activities only. This is not durable nor strategic in order to assure the society that all future engineers will be equipped with SD-competencies. Therefor the aim was to integrate CDE in the curriculum, at both UDSM and KTH, and not allowing any extra-curricular solutions. The complex nature of CDE as the object for the engineering education activity system change, as opposed to traditional education, calls for new approaches for development and implementation—the *how* of the change [22]. It is previously emphasized that ESD should be integrated with participatory approaches to make lasting changes where the ownership is in the hands of the key actors [8]. This involving approach is also supported in literature on general HE change [18,23,24,32]. With the PAR methodology this three-year project has been organized with the attempt to develop, practice and evaluate a collectively shared scenario of CDE at KTH and UDSM, and analyze the mechanisms and barriers for change with activity theory (AT). We argue that participatory design and research in the CDE implementation has been vital in assuring real change through contextual expertise and long-term ownership [32,46]. The PAR methodology has continuously forced the team members to gather, reflect and act based on the shared desired goal. The commitment, involvement and ownership is clearly revealed in the students', teachers' and challenge providers' reflections. Also, the clear shift between the two phases of implementation where the team members at UDSM took over the full implementation responsibility at their university is an important sign of this. Since there were not many prepared tools and artefacts for the implementation of CDE among any of the team members, all needed to contribute with their perspectives and ideas, which supported the maintenance of high ownership. This tells us that when an unfamiliar object is to be implemented in engineering education, participatory design is a valuable methodology, since it attends to the whole activity system, including the collective and social elements.

From the findings of the three-year CDE implementation project, the activity theory analysis tells that contradictions, or "ambiguity, surprise, interpretation, sense-making, and potential for change" [64] (p. 47), have been revealed and acted upon in the engineering education system when implementing CDE. Trust has been gained, where new dependencies have been supported with tools

and artefacts for dialogue in the increasingly complex education communities. These communities in the challenge-driven engineering education initiatives, as well as in many other community-based approaches, remove barriers between the university and the society on the outside. While the university through the implementation of CDE suggests that their students can contribute to sustainable development in society, the door is also opened for society's influence on higher education, in what is learned, taught and assessed. Through the lens of Activity Theory it is revealed that with the implementation of CDE a negotiation is initiated related to the norms and the values of what engineering education really is as well as what it should look like in the future [4]. Should we continue to value high scores on a pre-defined exam or should we work with open-ended and real-life tasks in society? These are two quite different positions, which provides with different answers to questions such as the balance between theory and practice, the meaning of learning for specific targets or for a lifelong learning, the level of teacher-centered vs student centered education, and the main critical contradiction in this project, namely the traditional pre-defined assessment framework against the open-ended. This tells us that when introducing a relevant and useful object such as the CDE, which is mutually co-created with colleagues and peers, the opportunities for the change to be lasting is mainly located in the norms and rules of the organization. Therefore we need to take into account that in order for the future of engineering education to grow in the sense of complexity, with inter and trans-disciplinary [19] as well as cross-national and intercontinental education activities [15], and lifelong learning [9], the conditions for change should be analyzed in the policy and vision of the institutions.

## 8. Conclusions from the Implementation of CDE at UDSM

In the light of the collectively gained experiences of the CDE implementation among teachers, students and societal stakeholders, the overall question for the study has been targeting what we can learn about participatory education change from an activity system perspective, when integrating CDE in engineering education, and how this can inform future steps for the integration of CDE. The conclusions are that the concept of sustainable development can be attractive, feasible and mutually understood among students and teachers, with the help of CDE due to its relevant and useful nature. Furthermore, participatory design and similar co-creative development processes are notably useful when implementing CDE, as they attend to social and cultural values among the participants which is crucial for a lasting ownership of something as complex as CDE. Finally, while many barriers for change will be removed by the teachers, students and the challenge providers, when implementing CDE, the norms and the rules of higher education will need to successively be in line with the intentions of the change, due to the open-ended nature of CDE.

**Author Contributions:** Conceptualization, A.-K.H., A.R., C.M., A.L., L.G., E.S., S.L. and N.M.; methodology, A.-K.H., A.L., N.M., S.L., C.M. and E.S.; methodology, A.-K.H.; validation, A.-K.H., A.R., C.M., A.L., L.G., E.S., S.L. and N.M.; formal analysis, A.-K.H., A.R. and A.L.; investigation, A.-K.H., A.L., N.M., S.L., C.M., L.G. and E.S.; resources A.-K.H.; data curation, A.-K.H., A.L., S.L., C.M. and E.S.; writing—original draft preparation, A.-K.H., A.R., A.L., L.G., C.M. and E.S.; writing—review and editing, A.-K.H. and A.R.; visualization, A.-K.H., E.S. and L.G.; supervision, A.L. and N.M.; project administration, A.-K.H., A.L., S.L., N.M., S.L., C.M. and E.S.; funding acquisition, A.-K.H., A.L. and N.M.

**Funding:** The development project was funded by STINT, The Swedish Foundation for International Cooperation.

**Acknowledgments:** We would like to acknowledge the importance of the support from STINT. Furthermore, creative discussions on two conferences (CDIO, 2018) and (EDUCON2018) gave valuable input. Also, to all students, teachers, external stakeholders involved in the CDE implementation, a warm thank you.

**Conflicts of Interest:** The authors declare no conflict of interest.

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
