# Peer review of "Mutual Capacity Building through North-South Collaboration Using Challenge-Driven Education"

_sustainability, doi:10.3390/su11247236_

Round 1
Reviewer 1 Report
Thank you for the interesting manuscript.
Points for improvement:
Check for faulty page breaks like in line 486 or 842 for example. Introduce a figure with a "flow chart" of all the case study phases and parts to inform the Reader more clearly. Make clear where citations like in lines 587 or 589 come from. More literature and references can be included. Delete the emtly lines in the reference list.Good luck for improvement!
Author Response
Dear reviewer,
Thanks for your valuable input. A flow chart is now included and called figure 1.
More referenceres are included, from 40 to 72.
Empty lines deleted, and clearer referring the quotes and explaining the questionnaire respondents.
Many thanks!
Reviewer 2 Report
The purpose of this research is to describe, evaluate and discuss a three years participatory implementation project of Challenge driven education (CDE) at the University of Dar Es Salam, UDSM, which has been carried out in collaboration with the Royal Institute of Technology, KTH in Stockholm. Although it is an interesting and meaningful article, the paper would be stronger if the following comments/suggestions are responded:
p.3 -- The author(s) need to take a more balanced view of higher education change; in other words, please explain more about the potential threats or challenges of it. p.1-3-Although the introduction provides a detailed insight into the context for this paper, more related work on engineering education of previous literature needs to be clarified. p.6 --. Please reconsider provide more details about the justification of using Participatory Action Research design and provide a figure of the stages of the PAR. p.11 –It would be better if the authors explained about the data analysis procedures clearly. Please clarify the data analysis processes. p.20 – The trustworthiness of this research and the limitation of the study need to be clarified.
Author Response
Dear reviewer,
Many thanks for your valuable input. The introduction and the background on higher education change is now filled with a more nuanced thickness, with more literature included.
The justification of PAR has now a more emphasized section.
Data analysis clarified, a new sub-chapter about this (Activity Theory). and the whole new methodology section brings forward clearer trustworthiness and drawbacks.
Many thanks!
Reviewer 3 Report
This article discusses an application of a participatory action-based research (PAR) approach for developing and implementing Challenge Driven Education (CDE) for Sustainable Development (ESD) at an University. Author’s aim is to créate a “collective reflection-in-action” and to answer this question: What can we learn about participatory education change when integrating CDE and Sustainable Development SD goals in engineering education and how can this inform future steps for integration of CDE?
The paper deals with an interesting problem and show and really interesting work but, in opinion of this reviewer, it should be improved.
The introduction and literature review (section 2) is really interesting, but It would be recommended that the authors make an effort to rewrite section 2 more concisely. References to the advancements on the topic are missing, and please explain how your article builds on these.
A list of abbreviations could make the document easier to read.
This paper appears to be an application case as the underlying methods do not appear to be new. It is advisable to completely describe the authors' methodology. Is the method described completely new, or a better version of an existing method? Authors could describe your methodology with a new section.
Please provide a brief description of the implementation. The author could also include a figure with all the steps, and highlight the new steps evaluated in this article. New material and method section?
It would be recommended that the authors make an effort to rewrite section 3 more concisely. Feedback is partially found in section 3 lines 538-539, but: How and when did Authors collect feedback from students, teachers and stakeholders? Are these data table 1 or table 2? Are S1-S14, T1-T4 or C1- C6 in table 1 or 2? Please clarify. Table 1 and 2 are not included in the paper (see lines 270, 281, 540….)
A statistical analysis of feedback results could be included. How have it impacted work, were the results better than in other methodologies or practical cases? Authors should indicate the advantages and disadvantages of your proposal compared to other
Please review the typographical problems.
Author Response
Dear reviewer,
Your review has been very valuable to this major revision.
The chapter 2 is now re-written with thicker material.
A specific chapter 3 is now included on Methodology, including a new section on Activity theory as framework for analysis in order to take the research to a new level. This is also reflected in the analysis section that comes after the findings, and also guides the discussion clearly as well as the conclusion. The research question has also been developed for this, to include the lens of activity theory.
A data and materials section is now included and the table 1 and 2 give the overview of the processes. when and where the data for the findings are gathered is clarified, as are the S1-S15 etc.
Typos and language should be fixed now.
Again, many many thanks, your review made this revision work really valuable and interesting.
Round 2
Reviewer 2 Report
I enjoy reading this novel and interesting manuscript. Thank you for sharing the research outcomes. However, some of the references are incompleted (i.e. page numbers or volume numbers are missing). If possible, please veried it before publication.
Author Response
Dear reviewer,
Many thanks for your nice comments. Also, all quotations are now referred to with page numbers, and the reference list is revised. Many many thanks.
Reviewer 3 Report
The authors have performed several proposed changes.
Now the contribution is more understandable, but please review the typographical problems (for example line 287-298) and review figure 3.
Author Response
Dear reviewer,
Many thanks for your reviews! Typos are now worked on, so line breaks and gaps are fixed. The figures are worked on. and the headings to figures and tables are according to template, and also referred to as supposed to in the text.
Thanks again!